# Proteomic Changes in Sarcoplasmic and Myofibrillar Proteins Associated with Color Stability of Ovine Muscle during Post-Mortem Storage

**DOI:** 10.3390/foods10122989

**Published:** 2021-12-03

**Authors:** Xiaoguang Gao, Dandan Zhao, Lin Wang, Yue Cui, Shijie Wang, Meng Lv, Fangbo Zang, Ruitong Dai

**Affiliations:** 1College of Food Science and Biology, Hebei University of Science and Technology, No. 26 Yuxiang Street, Yuhua District, Shijiazhuang 050000, China; gaoxiaoguang23@hotmail.com (X.G.); zdd6364@126.com (D.Z.); w592148022@163.com (L.W.); cuiy1223@hotmail.com (Y.C.); mrshjwang@163.com (S.W.); lmeng1126@163.com (M.L.); 15076005611@163.com (F.Z.); 2College of Food Science and Nutritional Engineering, China Agricultural University, No. 17 Qinghua East Road, Haidian District, Beijing 100083, China

**Keywords:** ovine, color stability, enzyme, proteomics, bioinformatics

## Abstract

The objective of this study was to investigate the proteomic characteristics for the sarcoplasmic and myofibrillar proteomes of *M. longissimus lumborum* (LL) and *M. psoasmajor* (PM) from *Small-tailed Han* Sheep. During post-mortem storage periods (1, 3, and 5 days), proteome analysis was applied to elucidate sarcoplasmic and myofibrillar protein changes in skeletal muscles with different color stability. Proteomic results revealed that the identified differentially abundant proteins were glycolytic enzymes, energy metabolism enzymes, chaperone proteins, and structural proteins. Through Pearson’s correlation analysis, a few of those identified proteins (Pyruvate kinase, Adenylate kinase isoenzyme 1, Creatine kinase M-type, and Carbonic anhydrase 3) were closely correlated to representative meat color parameters. Besides, bioinformatics analysis of differentially abundant proteins revealed that the proteins mainly participated in glycolysis and energy metabolism pathways. Some of these proteins may have the potential probability to be predictors of meat discoloration during post-mortem storage. Within the insight of proteomics, these results accumulated some basic theoretical understanding of the molecular mechanisms of meat discoloration.

## 1. Introduction

Among various sensory characteristics, the color of fresh meat is a critical quality factor, influencing the purchase decision of the consumers [1]. The defects of meat color have always been connected to spoilage and unwholesomeness [2]. Prevention of meat discoloration has always been a challenging task and long-term objective for scientific researchers [3,4,5]. Consequently, maintaining the “cherry-red” color of fresh meat is very critical for the meat industry.

The relative contents of the reduced deoxymyoglobin (DeoMb, purple), the oxygenated oxymyoglobin (OxyMb, bright red), and the oxidized metmyoglobin (MetMb, brown) determine the color of fresh meat [6]. The reason for meat discoloration is most commonly attributed to the formation of MetMb. Furthermore, the affecting factors of redox forms of myoglobin are formed by many intrinsic and extrinsic ones such as breed, muscle type, temperature, and oxygen partial pressure of storage [7,8]. Because of the distinct metabolic function and biochemical profile, *M. longissimus lumborum* (LL) and *M. psoas major* (PM) were considered to be typical color-stable and color-labile muscles, respectively, by previous researchers [9,10]. Thus, both of the skeletal muscles whose color stabilities (color-stable and color-labile) were opposing can be used as typical experimental subjects to elaborate the underlying biochemical mechanisms of post-mortem color stability [11]. The post-mortem changes in muscles constitute a very complicated biochemical processes influencing meat color stability, as skeletal muscle consist of proteins, lipids, and other biomacromolecules and micro-molecules [11,12]. Proteomics provides an efficient way, in post-genome era, to illustrate the post-mortem changes in meat quality development, including the field of color characteristics [13,14]. In the previous literature, some investigators explored a series of post-mortem proteomic changes in beef and pork [15,16,17,18]. However, data of physiological and biochemical mechanisms of ovine color stability during post-mortem storage has rarely been reported [7,19]. The proteomics for the variations of different ovine muscles (color-stable and color-labile) still need to be investigated. Therefore, the potential molecular mechanisms need further research. The objective of this study was to investigate the proteomic characteristics for the sarcoplasmic and myofibrillar proteome of *M. longissimus lumborum* (LL) and *M. psoasmajor* (PM) from *Small-tailed Han* Sheep during post-mortem storage.

## 2. Materials and Methods

### 2.1. Sample Preparation

The skeletal muscle samples were harvested from 6 male Small-tailed Han Sheep, slaughtered at 8 months, with a mean carcass weight of 15.6 ± 0.2 kg. There were 2 typical types of skeletal muscles (*M. longissimus lumborum* (LL) and *M. psoas major* (PM)) that were excised (30 min after slaughter) from both sides of each carcass (*n* = 6 carcasses). The whole process of slaughter followed the industrial practice. The animals received humanely nonpainful manipulations at the termination of the procedure, without regaining consciousness. Muscles samples were cut along the direction perpendicular to the muscle fibers. For each sample (corresponding to 1 carcass and 1 type of muscle), 6 equal-weight (30 g, trimmed free of connective tissue and fat) pieces were randomly assigned to 12 replications [*n* = 6, (6 muscles per type × 2 sides × 3 sections per muscle)/3 days]. 

The samples were wrapped individually with polyethylene films (350–400 cm^3^m^−2^h^−1^atm^−1^, Mitsui Chemical, Japan) and put in a refrigerator for a storage time of 5 days (4 ± 1 °C). Muscles were taken out for analyses at day 1, day 3, and day 5. At each time point (day 1, day 3, and day 5), instrumental color parameters and metmyoglobin reducing activity (MRA) of the muscles were measured. Sarcoplasmic and myofibrillar proteins were extracted for comparing the change in the early stage of storage (day 1) and the late stage of storage (day 5). At the storage time of day 3, proteins were not extracted accordingly. At the end of storage period, myofibrillar proteins were extracted for the comparison of proteomic difference between muscle types (LL and PM).

### 2.2. Instrumental Color

All the measurements of color parameters were determined on the surface of each sample. The instrumental color parameters were measured by utilizing a Minolta chromameter (CR-400, Minolta Inc., Osaka, Japan), expressing as CIE Lab lightness (*L**), redness (*a**), and yellowness (*b**) with D_65_ standard illuminant. Accordingly, the parameters were calibrated through the white and black reference standards. The observer angle and measurement area were 2° and 8 mm, respectively. Hue angle (h°) and chroma (*C**) were determined by the following formula: h° = [ATAN(*b**/*a**) × (180/π)], *C** = [(*a**^2^ + *b**^2^)^0.5^] [20].

LL and PM samples (5 g of each sample) were collected and homogenized for 10 s in a phosphate buffer (25 mL, pH 6.8, 0 °C, 40 mM) by using a homogenizer (F6-10, Fluko, Shanghai, China). The homogenate solution was placed in 4 °C for an hour and centrifuged (4500× *g*, 4 °C). Supernatant fluid was collected and filtered (Whatman-No.1 filter paper). The absorbance of the filtrate was tested with a spectrometer (wavelength of 503, 525, 557, 582 nm; TU-1810, PERSEE, Beijing, China). The proportion of myoglobin redox forms was calculated as follows [21]: [DeoMb] = C_DeoMb_ ÷ C_Mb_ = − 0.543R_1_ + 1.594R _2_ + 0.552R_3_ − 1.3290
[OxyMb] = C_OxyMb_ ÷ C_Mb_ = 0.722R_1_ − 1.432R_2_ − 1.659R_3_ + 2.599
[MetMb] = C_MetMb_ ÷ C_Mb_ = − 0.159R_1_ − 0.085R_2_ + 1.262R_3_ − 0.520(1)
where R_1_ = A_582_ ÷ A_525_; R_2_ = A_557_ ÷ A_525_; R_3_ = A_503_ ÷ A_525_.

### 2.3. Metmyoglobin Reducing Activity

The metmyoglobin reducing activity (MRA) was determined through spectrophotography and based on Mikkelsen et al. [22], including the following experimental equipment: spectrometer (TU-1810, PERSEE, Beijing, China), refrigeration centrifuge (3K15, SIGMA, Hamburg, Germany) and homogenizer (F6-10, Fluko, Shanghai, China). Muscle sample (12 g, connective tissue and fat removed) was homogenized (20 mL, 2.0 mM phosphate buffer, pH 7.0) for 30 s. The homogenate was centrifuged (35,000× *g*, 4 °C) for 30 min. The supernatant fluid was collected and filtered (Whatman-No.1 filter paper) for further removal of fat. 

K_3_Fe(CN)_6_ was used to oxidize OxyMb into MetMb. The solution was dialyzed against phosphate buffer (2.0 mM, pH 7.0, 4 °C, 14,000 Mw cut-off) and centrifuged for 20 min (15,000× *g*, 4 °C). The supernatant was adjusted to volume of 20 mL through phosphate buffer (2.0 mM, pH 7.0). The standard assay (pH 6.4, 25 °C) was the mixed solution of NADH (0.1 mL, 2.0 mM, used for initiating the reaction), K_4_[Fe(CN)_6_] (0.1 mL, 3.0 mM), EDTA (0.1 mL, 5.0 mM), Mb Fe(III) (0.2 mL of 0.75 mM within 2.0 mM phosphate buffer, pH 7.0), phosphate buffer (0.1 mL, 50 mM, pH 7.0), deionized water (0.1 mL) and muscle extract (0.3 mL). The blank contained all chemicals except for NADH (replace NADH with water, no reaction). MRA was followed the changes in absorbance (580 nm) and determined as nmol MetMb reduced (min^−1^ g^−1^).

### 2.4. Sarcoplasmic and Myofibrillar Protein Extraction

Sarcoplasmic and myofibrillar protein extraction methods were based on Sayd et al. [23] and Kim et al. [24] with a minor modification, respectively. Each muscle sample (3 g) was homogenized with extraction buffer (30 mL, 2 mM EDTA, 40 mM Tris, pH 7.0, 4 °C) containing a protease inhibitors cocktail by utilizing a homogenizer (F6-10, Fluko, Shanghai, China). The sample was centrifuged (10 min, 10,000× *g*, 4 °C) and the supernatant (sarcoplasmic proteins) was collected for further experiments (stored in −80 °C). The precipitate of each sample was incubated for 40 min in 2 mL of extraction buffer (1% pH 3–10 bio-lyte ampholytes (Bio-Rad, Hercules, CA, USA), 2 M thio-urea, 8 M urea, 2% CHAPS (*w*/*v*), 65 mM DTT, with protease inhibitors cocktail). The sample was centrifuged (40,000× *g*, 4 °C) for 60 min and the supernatant (myofibrillar proteins) was stored at −80 °C for further experiments. 

### 2.5. Two-Dimensional Electrophoresis and Gel Image Analysis

The sarcoplasmic and myofibrillar proteome (800 μg, respectively) was included in the Bio-Rad buffer (8 M urea, 2% CHAPS (*w*/*v*), 50 mM DTT, 0.3% carrier ampholyte (*v*/*v*), bromophenol blue). The samples were loaded on the IPG strips (immobilized pH gradient, 24 cm, pH 3–10, Bio-Rad). The PROTEAN II XL and PROTEAN IEF (isoelectric focusing) cell system (Bio-Rad, Hercules, CA, USA) performed in the first and second dimension electrophoreses, respectively. 

In the first dimension, IEF was subjected to passive rehydration (16 h) and then rapid voltage ramping (80,000 V h) was applied. In the second dimension, proteins were resolved on 12% SDS-PAGE gels, which were stained in Coomassie Brilliant Blue (48 h). Gel images were scanned by an image scanner (ImageScanner III, GE Healthcare, Branford, CT, USA). ImageMaster 2D Platinum software (6.0 version, GE Healthcare, Branford, CT, USA) was used to analyze the scanning images. Through expressing the relative number of each spot as the ratio of the number of single spots to the total number of valid spots, the detected and matched spots were normalized. The mean values of gels for each sample and spot were calculated in triplicate. A spot was considered as differential proteins when it came to 5% statistical significance (*p* < 0.05) in one-way ANOVA.

### 2.6. Identification of Protein Spots by Mass Spectrometry 

The protein spots were carefully excised from gels and then de-stained for 30 min in wash buffer (100 μL, 25 mM NH_4_HCO_3_/50% acetonitrile (*v*/*v*)). Washed-out gel-spots (dehydrated in 100% acetonitrile) were dried completely with a centrifuge (Vacufuge plus, Eppendorf, Hamburg, Germany) and then incubated in 15 ng/μL trypsin and 25 mM NH_4_HCO_3_ (37 °C, 16 h). The peptides were incubated in trifluoroacetic acid (20 μL, 0.1% (*v*/*v*), 37 °C, 40 min) after digestion. The above extraction procedures were repeated by using 50% acetonitrile/0.1% trifluoroacetic acid (*v*/*v*). Sediments were washed in trifluoroacetic acid and then vacuum freeze-dried for further analysis. The proteins were identified by using AUTOFLEX II TOF-TOF mass spectrometer (autoflex™ speed, Bruker Daltonik, Bremen, Germany). Samples in 1 μL of buffer (50% acetonitrile and 5 mg/mL α-CHCA in 0.1% trifluoroacetic acid) were loaded on plate (AnchorChip, 384-MPT).

Each crystallized sample was washed by using 0.1% trifluoroacetic acid for removing salt ions. Protein identification was performed by peptide mass fingerprinting (PMF), searching the Mascot (2.2 version, Matrix Science, London, UK), and matched with a sheep (*Ovis aries*) family in the Uniprot database (https://www.uniprot.org/ (accessed on 8 October 2021)), correspondingly.

### 2.7. Statistical Analysis

Proportions of MetMb, OxyMb, DeoMb, and instrumental color attributes (*L**, *a**, *b** and MRA) were analyzed separately. Random terms for all models included the carcass and the processing day. For color models, an additional term for side nested within the carcass was added to the random model to account for the repetition of measures [25]. The correlation between meat color attributes and differentially abundant proteins was analyzed by Pearson’s correlation. The mixed models, which were prediction models for muscle color traits, were based on Starkey et al.

Data were expressed as mean ± SE (standard error). The GLM (general linear model) procedure was used to analyze the instrumental color and MRA data with the SAS 8.2 software (Institute Inc., Cary, NC, USA). Means and the correlation data were analyzed by one-way ANOVA by using the SPSS 20.0 software (IBM Inc., Chicago, IL, USA) with the 5% (*p* < 0.05) level of statistical significance difference. 

Protein–protein interaction (PPI), gene ontology (GO), and Kyoto encyclopedia of genes and genomes (KEGG) were applied to characterize the functional information of the identified proteins. The protein–protein interaction was analyzed by String 10.0. Bioinformatics analysis was performed through DAVID Bioinformatics Resources 6.7. The species of *Ovis aries* was selected.

## 3. Results and Discussion

### 3.1. Instrumental Color and MRA

The instrumental color parameters of skeletal muscles (*M. longissimus lumborum* (LL) and *M. psoas major* (PM)) are presented in Table 1. There was an insignificant (*p* > 0.05) decrease trend in *L**-value. Besides, no significant difference (*p* > 0.05) was found between these two kinds of skeletal muscles within storage time points in *L**-value. Although some previous studies [26] showed that the variation in *L**-value were very subtle. Besides being a color parameter, the *L**-value is also considered an indicator of the water-holding capacity of meat [11]. During ageing in ovine muscle, structural changes of proteins were considered one of the causes of the change in lightness (*L**-value) [27]. 

For the other color parameters, there were decreasing trends in redness (*a**-value), yellowness (*b**-value), and chroma (*C**-value) of both kinds of muscles. Meanwhile, hue angle (h°-value) exhibited increasing trends during 5 days of post-mortem storage. Those results were consistent with studies in the previous literature [26,28,29]. In an associated study [17], the relative content of OxyMb, MetMb, and DeoMb, on the value of the color parameters of minced pork loin, were evaluated. The color parameters were associated with myoglobin redox forms. Our results (Figure 1) were partially in agreement with Karamucki et al. [17]. For instance, an increase in OxyMb contributes most greatly to an increase in redness (*a**-value), yellowness (*b**-value), and chroma (*C**-value). Chroma (*C**-value) was deemed as an indicator for vividness of color [30]. Compared single color coordinate, hue angle (h°-value), showed more realistic perspectives on meat discoloration [8]. In the literature, muscles with unstable color often go along with greater hue angle (h°-value) [10]. In this study, compared with PM muscle, LL showed lower (*p* < 0.05) hue angle (h°-value) values, indicating a greater color stability.

MRA (metmyoglobin reducing activity) decreased with storage time. LL showed greater metmyoglobin-reducing activity than PM (*p* < 0.05, Table 1). Muscles with more color stability had greater ability to reduce metmyoglobin into reduction state. In practice, the loss of reducing activity was influenced by many cofactors, such as the decline of pH, the depletion of substrates, the loss of respiration enzymes or functional properties, and the structural integrity of mitochondria during post-mortem storage of meat [29,31]. MRA gradually decreased, partially due to depletion of the NADH pool and reduction in material consumption [32]. Consequently, LL muscle displayed greater color stability than PM during storage period.

### 3.2. Sarcoplasmic and Myofibrillar Proteome Analysis

The differentially abundant (*p* < 0.05) proteins, which were identified by two-dimensional electrophoresis image analysis of ovine muscles (*M. longissimus lumborum* (LL, color-stable) and *M. psoas major* (PM, color-labile)), were presented in Figure 2. Differential abundance changes of spots were at least twofold (*p* < 0.05). The identification and related information ware shown in Table 2. The identified proteins were matched with the sheep (*Ovis aries*) family in the Uniprot database. According to physiological function in metabolism, the differentially abundant sarcoplasmic and myofibrillar proteins were classified into three categories: enzymes, chaperone proteins, and structural proteins.

Please note: The early stage of storage (day 1) of myofibrillar proteins gel images were not shown, as there was little differentially abundant proteins between LL and PM muscles at day 1. 

### 3.3. Correlation of Differentially Abundant Proteins with Meat Color Attributes

The correlation data between meat color attributes and differentially abundant sarcoplasmic proteins are presented in Table 3. In this research, the proteomic changes of proteins may affect the color traits of ovine muscle. Therefore, the method of Pearson’s Spearman’s correlation was used to infer the possible correlation between meat color attributes and differentially abundant sarcoplasmic proteins.

During post-mortem storage, six proteins were correlated with MRA and instrumental color data in *M. longissimus lumborum* muscle (Table 3). There was a significant correlation between redness (*a**-value) and adenylate kinase isoenzyme 1 or creatine kinase M-type (r = 0.950, 0.955; *p* < 0.01). Contrarily, a negative relationship was found between hue angle (h°-value) and adenylate kinase isoenzyme 1 or creatine kinase M-type (r = −0.956, 0.957; *p* < 0.01). Chroma (*C**-value) also has been found a positive correlation with carbonic anhydrase 3, adenylate kinase isoenzyme 1 or creatine kinase M-type (r = 0.857, 0.876, 0.877; *p* < 0.05). MRA of LL and PM muscles were positively correlated with adenylate kinase isoenzyme 1 and creatine kinase M-type (r = 0.876, 0.889; *p* < 0.05). 

Those proteins (overabundant in color-stable muscles), such as fructose-bisphosphate aldolase [16], glyceraldehyde-3-phosphate dehydrogenase [33], adenylate kinase isoenzyme 1, pyruvate kinase [34], and creatine kinase M-type [16,29], exhibited positive correlations with several meat color parameters (*a**-value, *b**-value, and *C**-value), associating them with stability of meat color. In addition, Joseph et al. [12] observed that some sarcoplasmic proteins (antioxidant) showed positive correlations with *a**-values and MRA. Those results were partially in agreement with present study.

### 3.4. Functional Roles of Differentially Abundant Proteins and Their Relevance to Color Stability

#### 3.4.1. Glycolytic Enzymes

Four sarcoplasmic proteins (glyceraldehyde-3-phosphate dehydrogenase, fructose-bisphosphate aldolase, pyruvate kinase, enolase 2), which belong to glycolytic enzymes, were over-abundant in LL group (color-stable muscle, Table 2). 

Fructose-bisphosphate aldolase catalyzes the reaction that splits the fructose 1,6-bisphosphate into glyceraldehyde 3-phosphate, and dihydroxyacetone phosphate [35]. During the early stage of storage, fructose-bisphosphate aldolase was over-abundant in LL muscle. Nevertheless, an opposite result was observed during late stage of storage (Table 2). Fructose-bisphosphate aldolase was closely related to fast-twitch fiber displaying a higher glycolysis metabolic activity [2]. The down-regulated trend of fructose-bisphosphate aldolase involved in glycolytic metabolism indicated that glycolytic activity was highly active in PM muscle which exhibited lower redness (*a**-value) and color stability [34].

The glycolysis pathway continues with the conversion of glyceraldehyde 3-phosphate into dihydroxyacetone phosphate, the reaction of which is catalyzed by glyceraldehyde 3-phosphate dehydrogenase. The reaction is NAD-dependent, resulting in the production of pyruvate and NADH [34,35]. Pyruvate kinase catalyzes conversion of phosphoenolpyruvate (PEP) and ADP into ATP and pyruvate, which is one of the main rate-limiting enzymes in glycolysis [35,36]. Enolase (also called muscle-specific enolase) catalyzes the conversion of 2-phosphoglycerate to PEP [37]. The three glycolytic enzymes mentioned above were over-abundant in LL group (Table 2). Glyceraldehyde 3-phosphate dehydrogenase and pyruvate kinase exhibited a positive correlation with MRA (Table 3). Some researchers have discussed the differences between color-stable and color-labile muscles in the literature. Several glycolytic enzymes (β-enolase, pyruvate kinase M2, and glyceraldehyde-3-phosphate dehydrogenase), which were over abundant in *Longissimus lumborum* (color-stable muscle), were found to be positively correlated with *a**-values [12,34]. The over-abundance of the enzymes mentioned above could result in higher metabolic activity of glycolytic pathway [34]. This stimulated biological process could accelerate the NADH and pyruvate production. Furthermore, pyruvate is one of the mitochondrial substrates that promote the regeneration of NADH [38]. Besides, NADH supplied by glycolysis is necessary for the NADH–cytochrome b_5_ reductase system of mitochondria [39]. The NADH–cytochrome b_5_ reductase system was deemed as an important electron transport-mediated reduction system for metmyoglobin, whose mechanism has been expounded by some researchers [7]. Researchers [30] considered NADH a critical component of MRA, whose metabolic function plays a key role in reducing metmyoglobin accumulation [19]. In fact, a direct relationship was found between color stability and MRA, in that muscles with enzyme activity (MRA) in higher had greater color stability [7]. Consequently, NADH may play a key role in MetMb enzymatic reduction, which further influences color stability of meat [2].

In this study, Enolase 2 was overabundant in the LL group (color-stable muscle, Table 2). Enolase, one of the key enzymes in glycolysis, catalyzes the formation of phosphoenolpyruvate from 2-phosphoglycerate, leading to an increased rate of glycolysis [40]. Previous literature findings showed that enolase 1 was related to meat color development and quality variation [23,41,42]. In some proteome research, the profile of muscles of different color stability was compared, revealing the correlation between enolase and attributes of meat color [12,32]. The findings in proteomic analysis indicated that enolase was a key enzyme associated with the color of meat.

#### 3.4.2. Energy Metabolism Enzymes

Adenylate kinase isoenzyme 1 was overabundant and positively correlated with MRA and chroma (*C**-value) in the color-stable LL muscle (Table 2 and Table 3). Adenylate kinase (also known as myokinase), a reversible enzyme, plays a role in adenine synthesis and energy metabolism [43]. It provides a way to catalyze the conversion of ADP to ATP and convert AMP to ADP in the cytoplasm [44]. There were different findings within adenylate kinase in the previous associated muscle proteomic studies. Canto et al. [34] noted that the exact mechanism of how adenylate kinase affects color stability remains unclear. Wu et al. [2] found a positive relationship between this enzyme and MRA. It was considered to be a possible potential predictor for meat color stability of M. *longissimuss lumborum*. Thus, the exact function mechanism of adenylate kinase in proteomics of muscle color stability is waiting for further investigation.

Creatine kinase M-type was over-abundant in LL muscle (Table 2), demonstrating positive relationships (r = 0.702, 0.955; *p* < 0.05) with redness (*a**-value) and MRA (Table 3). Creatine kinase M-type was critical in maintenance of ATP-ADP level, reversibly catalyzing the mutual transformation of phosphate between ATP and phosphocreatine [16]. Creatine kinase was degraded during the early post-mortem stages due to the depletion of ATP [42]. Findings from previous literature showed that the denaturation of creatine kinase and several other related proteins were considered to be related to the paler colored (lower *a**-value) *longissimus lumborum* muscle [45]. The present study was in agreement with a previous study, which found that *longissimus lumborum* muscle that creatine kinase M-type was positively correlated (*p* < 0.05) with *a**-values and MRA [16]. Creatine can be used as a natural selective antioxidant because of its ability to scavenge oxygen free radicals [46,47]. 

Creatine kinase M-type, which is relatively overabundant, can increase creatine levels and minimize myoglobin oxidation, which leads to a lower formation of metmyoglobin and enhanced color stability in color-stable muscles [34].

The functions of carbonic anhydrase 3 were considered as an oxyradical scavenger. In skeletal muscle, this enzyme prevents cells for oxidative damage [48]. In agreement with our results, Yu et al. found that carbonic anhydrase 3 was overabundant in *M. longissimus lumborum* (color-stable muscle) compared with *M. psoas major* (color-labile muscle) [11]. Moreover, we also found this enzyme was correlated with redness (*a**-value) and hue angle (h°-value) (r =0.702, −0.543; *p* < 0.05). In practice, an increased glycolysis and oxidative stress can be indicated by carbonic anhydrase 3 in skeletal muscle [49]. In color-stable muscles, the overabundance of this enzyme can increase glycolysis levels and minimize myoglobin oxidation, possibly resulting in lower redness (*a**-value) and stabilized color attributes [5].

#### 3.4.3. Chaperone Proteins and Structural Proteins

One of the physiological functions of chaperone protein is preventing protein from denaturation [23]. Myoglobin and meat color stability can be compromised by protein denaturation in the process of transformation from muscle to meat [12]. The major chaperone proteins and structural proteins identified were heat shock protein family B, actin (alpha 1), myosin light chain 1, serum albumin, and carbonic anhydrase 3.

The heat shock protein family B (HSP beta-1, also known as HSP27) belongs to chaperone proteins. HSP27 is involved in small heat shock protein family which can prevent structure damage or degradation of proteins in muscle cells and promotes the survival of cell under physiological stress [50,51]. In pig muscle, lower abundance of HSP27 was related to the development of lighter color and PSE zones in semimembranosus [23,52]. Higher abundance of HSP27 was found in beef which was lower in *a** and *L** values compared with their counterparts. [24]

The physiological function of myosin and actin are mainly related to contractile properties of muscle [43]. During muscle contraction, the function of alpha actin is associated with ATP binding. [53]. Actin alpha has been identified as related to the color of meat in some previous proteomic analyses [4,54]. In a similar study, alpha actin and myosin light chain 1, identified in porcine longissimus muscle, were related to *L** value. In the study of Gagaoua et al. [15], actin alpha was correlated with all color coordinates. This protein and several structural proteins were deemed as biomarkers for meat color and of great interest in discriminating between beef color classes. Thus, the functional role of structural proteins in meat color proteomics was important. 

### 3.5. Bioinformatics Analysis of Differentially Abundant Proteins

The analytical method of bioinformatics has been applied to the related research area of meat color investigation to explore the biological processes and molecular functions of proteins [2,41,55]. Bioinformatics analysis of differentially abundant proteins between muscle types (color-stable and color-labile) provides an efficient tool for deeper understanding of the molecular mechanisms of meat discoloration, based on the proteomic results. In this study, Protein–protein interaction (PPI), gene ontology (GO), and the Kyoto encyclopedia of genes and genomes (KEGG) analyses were applied to characterize the functional information of the identified proteins (DAVID Bioinformatics Resources 6.7).

#### Protein–Protein Interactions

The protein–protein interactions were illustrated in Figure 3 and assessed by String 10.0. There are nodes and edges networks, representing the identified proteins and functional annotation of protein–protein interactions with different colors. The local clustering coefficient and PPI enrichment *p*-value were 0.49 and 1.81 × 10^−8^ respectively.

The highest number of proteins that interacted strongly with each other were implicated in glycolysis pathway (ALDOB, GAPDH, PKM, and ENO2). Four proteins with close range were involved in energy metabolic process or structural protein (AK1, CKM, CA3, and ACTA1). Furthermore, ALB and HSPB11 were exhibited further distance with the other proteins in the network.

The nodes are differentially abundant proteins (colored nodes—query proteins and first shell of interactors; white nodes—second shell of interactors; empty nodes—proteins of unknown 3D structure; filled nodes—some 3D structure is known or predicted) in Ovis aries database and connection lines are the predicted functional annotations (known interactions: blue—from curated databases; purple—experimentally determined; and predicted interactions: green—gene neighborhood; red—gene fusions; dark blue—gene co-occurrence; and others: yellow—text mining; black—co-expression; light blue—protein homology).

## 4. Conclusions

In the present research, the post-mortem proteome profiles associated with the typical color-stable (*M. longissimus lumborum*) and color-labile (*M. psoasmajor*) muscles of *Small-tailed Han* Sheep were studied and compared. The results of differential proteome indicated that the identified proteins were glycolytic enzymes, energy metabolism enzymes, chaperone proteins, structural proteins, and chaperone and binding proteins. Several proteins were demonstrated to be related to post-mortem discoloration through the methods of Pearson’s correlation and multiple linear regression model analysis between differentially abundant proteins and meat color indices.

Bioinformatics analysis showed that these proteins (pyruvate kinase, fructose-bisphosphate aldolase, enolase 2, and glyceraldehyde-3-phosphate dehydrogenase) mainly involved in the glycolysis pathway. The relationship between post-mortem proteome profiles and discoloration could play a key role in explaining the different color stabilities within muscles. However, further research is needed to confirm if the specific differentially abundant proteins have the potential probability of being predictors for meat discoloration during post-mortem storage.

## Figures and Tables

**Figure 1 foods-10-02989-f001:**
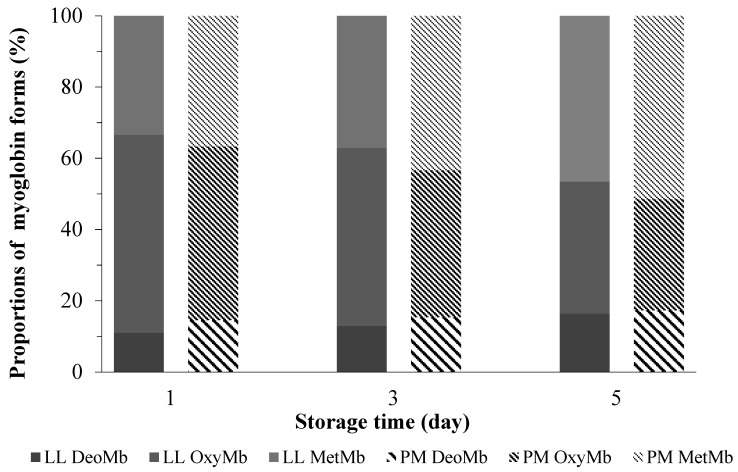
Changes in the relative proportions of MetMb, OxyMb, and DeoMb in ovine *M. longissimus lumborum* (LL) and *M. psoas major* (PM) muscles during postmortem storage (4 ± 1 °C).

**Figure 2 foods-10-02989-f002:**
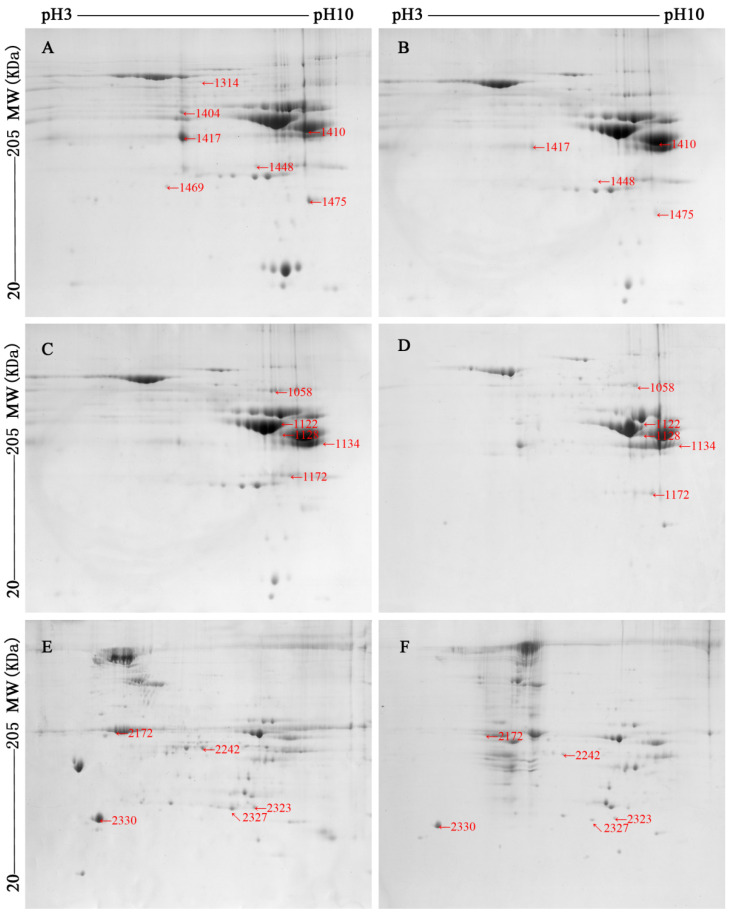
Two-dimensional gel images (pH range between 3 and 10 and molecular weight from about 20 to 205 kDa) of sarcoplasmic ((**A**,**B**), the early of storage, day 1; (**C**,**D**), the late stage of storage, day 5) and myofibrillar ((**E**,**F**), the late stage of storage, day 5) proteome from ovine *M. longissimus lumborum* (LL) and *M. psoas major* (PM) muscles.

**Figure 3 foods-10-02989-f003:**
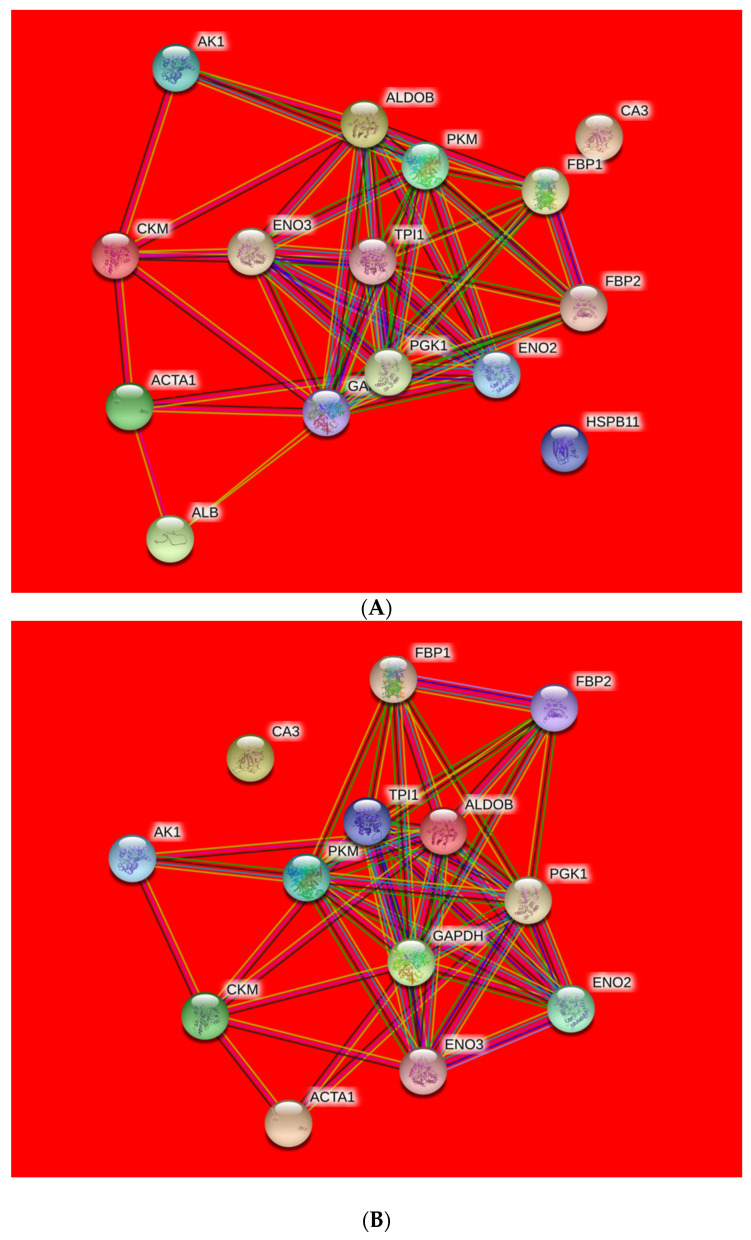
Protein–protein interaction (PPI) network of differentially expressed proteins from ovine *M. longissimus lumborum* (**A**) and *M. psoas major* (**B**) muscles.

**Table 1 foods-10-02989-t001:** Instrumental color attributes of ovine *M. longissimus lumborum* (LL) and *M. psoas major* (PM) muscles at 1, 3, and 5 days of postmortem storage (4 °C).

Attribute	Muscle		Storage Time (Day)		SEM
		1	3	5	
*L**	LL	44.25 ax	43.61 ax	37.83 by	0.403
	PM	44.81 bx	46.75 ax	44.70 bx	0.472
*a**	LL	21.41 ax	17.20 bx	13.62 cx	0.35
	PM	16.96 ay	12.98 by	10.59 cy	0.306
*b**	LL	15.57 ay	15.02 ax	12.72 by	0.262
	PM	17.33 ax	15.88 bx	13.28 cx	0.261
*C**	LL	26.47 ax	22.83 bx	18.64 cx	0.423
	PM	24.25 ay	20.51 by	16.99 cy	0.374
h°	LL	36.03 cy	41.12 by	43.06 ay	0.322
	PM	45.62 bx	50.74 ax	51.43 ax	0.372
MRA	LL	0.210 ax	0.145 bx	0.067 cx	0.003
	PM	0.170 ay	0.121 by	0.041 cy	0.004

Means in rows with different superscripts (a–c) are different (*p* < 0.05) and means in column for LL and PM for each storage time and attribute with different superscripts (x–z) are different (*p* < 0.05). (a) SEM, standard error of the mean. (b) *C** (Chroma) = (*a**^2^ + *b**^2^)^0.5^. (c) h° (Hue angle) = ATAN(*b**/*a**) × (180/π).

**Table 2 foods-10-02989-t002:** Differentially abundant proteins in ovine *M. longissimus lumborum* (LL) and *M. psoas major* (PM) muscles at day 1 or day 5 postmortem storage (4 °C).

Spot. *^(a)^*	Protein Name *^(b)^*	Uniprot ID *^(b)^*	Gene Names	Protein Score	Mw/pI *^(c)^*	Matched Peptides *^(d)^*	Sequence Coverage (%) *^(d)^*	Overabundant in Muscle *^(^**^e)^*
	*Sarcoplasmic**proteins* (The early stage of storage, day 1, corresponding Figure 2A,B)							
1314	Serum albumin	P14639	ALB	153	71,139/5.80	15	43	LD
1404	Creatine kinase M-type	W5PJ69	CKM	127	43,213/6.66	15	40	LD
1410	Fructose-bisphosphate aldolase	W5PCA0	ALDOB	96	41,555/7.57	12	34	LD
1417	Glyceraldehyde-3-phosphate dehydrogenase	W5PDG3	GAPDH	111	36,110/8.51	11	40	LD
1448	Carbonic anhydrase 3	W5PUC1	CA3	83	29,726/7.70	6	22	LD
1469	Heat shock protein family B	W5P9U1	HSPB11	75	15,737/5.08	5	34	LD
1475	Adenylate kinase isoenzyme 1	C5IJA8	AK1	172	21,750/8.40	15	62	LD
	*Sarcoplasmic**proteins* (The late stage of storage, day 5, corresponding Figure 2C,D)							
1058	Pyruvate kinase	W5QC41	PKM	147	58,551/7.24	20	40	LD
1122	Creatine kinase M-type	W5PJ69	CKM	109	43,213/6.66	14	37	LD
1128	Fructose-bisphosphate aldolase	W5PCA0	ALDOB	87	41,555/7.57	12	36	PM
1134	Glyceraldehyde-3-phosphate dehydrogenase	Q28554	GAPDH	95	36,110/8.51	13	39	LD
1172	Adenylate kinase isoenzyme 1	C5IJA8	AK1	147	21,750/8.40	14	64	LD
	*Myofibrillar proteins* (The late stage of storage, day 5, corresponding Figure 2E,F)							
2172	Actin, alpha 1	W5NYJ1	ACTA1	70	42,338/5.23	11	25	LD
2242	Enolase 2	W5P5C0	ENO2	117	47,382/7.60	14	37	LD
2323	Fructose-bisphosphate aldolase	W5PCA0	ALDOB	62	41,555/7.57	7	26	LD
2327	Creatine kinase M-type	W5PJ69	CKM	104	43,213/6.66	16	32	LD
2330	Myosin light chain 1	A0A0H3V7A0	MYL1B	73	20,950/4.95	9	58	LD

*^(a)^* The numbered spots in gel image (Figure 2). *^(b)^* Protein names and accession numbers were taken from the Uniprot database (http://www.uniprot.org (accessed on 8 October 2021)). *^(c)^* Theoretical protein mass (Mw; kDa) and isoelectric pH (pI). *^(d)^* Number of peptides that matched the protein sequence and total percentage of sequence coverage. *^(e)^* Muscle with greater abundance of the protein significance level was indicated (*p* < 0.05).

**Table 3 foods-10-02989-t003:** Correlation coefficients (Pearson and Spearman) of differentially abundant proteins in ovine *M. longissimus lumborum* (LL) with color attributes (*n* = 6).

Protein Name	Trait *^(a)^*	Pearson’s Correlation Coefficient *^(b)^*	Spearman’s Correlation Coefficient *^(b)^*
Fructose-bisphosphate aldolase (ALDOA)	*a**	0.126	0.086
	*C**	0.179	0.257
	h°	−0.061	0.086
	MRA	0.145	0.143
Gyceraldehyde-3-phosphate dehydrogenase (GAPDH)	*a**	0.127	0.314
	*C**	0.072	0.371
	h°	−0.139	−0.143
	MRA	0.232	0.486
Carbonic anhydrase 3 (CA3)	*a**	0.702	0.829 *
	*C**	0.857 *	0.771
	h°	−0.543	−0.771
	MRA	0.346	0.200
Adenylate kinase isoenzyme 1 (AK1)	*a**	0.950 **	0.771
	*C**	0.876 *	0.829 *
	h°	−0.956 **	−0.714
	MRA	0.876 *	0.657
Pyruvate kinase (PKM)	*a**	0.575	0.543
	*C**	0.416	0.543
	h°	−0.644	−0.429
	MRA	0.653	0.657
Creatine kinase M-type (CKM)	*a**	0.955 **	0.886 *
	*C**	0.877 *	0.886 *
	h°	−0.957 **	−0.878 *
	MRA	0.889 *	0.771

*^(a)^ C** means chroma; h° means hue angle. *^(b)^* * means significant at *p* < 0.05, ** means significant at *p* < 0.01.

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
