# Peer review of "Proteomic Changes in Sarcoplasmic and Myofibrillar Proteins Associated with Color Stability of Ovine Muscle during Post-Mortem Storage"

_foods, 2021, doi:10.3390/foods10122989_

Round 1

Reviewer 1 Report

Proteomic Changes in Sarcoplasmic and Myofibrillar Proteins associated with Colour Stability of Ovine Muscle during Post-mortem Storage

This article represents a significant contribution to scientific knowledge about the proteomic characteristics for the sarcoplasmic and myofibrillar proteome of two lamb muscles (M. longissimus lumborum and M. psoas major) and their relationships with muscle colour-stable and colour-labile.

The article is well organized and written clearly. Despite this, some improvements can be made to make the text clearer. The text addresses the subject with scientific correction. Tables and figures are relevant, but some corrections are needed to improve accuracy and clarity. The material and methods are clearly described, which will allow a perfect understanding by other researchers. The results are well discussed with the existing knowledge on the subject. Finally, the results support the conclusions.

Some detailed comments below:

L30 always been connected “change with” have always been connected

L33 for meat industry “change with” for the meat industry

L49 development including the development, including the

L60-61The samples of skeletal muscle were harvested from the Small-tailed Han Sheep (male) were slaughtered at the age of 8 months and mean carcass weight of 15.62 kg “change with” The skeletal muscle samples were harvested from 6 male Small-tailed Han Sheep slaughtered at eight months with a mean carcass weight of 15.6± kg. Please introduce the standard deviation of the carcass weight.

L63-64. Improve the description text of the slaughter procedure and introduce a sentence related with animal welfare.

L64 slaughter was following the industrial “change with” slaughter followed the industrial

L70 and stored in a in a refrigerator for storage time of five days “change with” and put in a refrigerator for a storage time of five days

L71 Muscles were taken out to analyse at day 1, day 3 and day 5 for further experiments. “change with” Muscles were taken out to analyse at day 1, day 3 and day 5.

L76 Please check the word accordingly. Please rewrite this sentence.

L79 determined for the surface of each “change with” determined on the surface of each

L91 Absorbance of filtrate was tested “change with” The absorbance of the filtrate was tested

L99 MRA (metmyoglobin reducing activity) “change with” The metmyoglobin reducing activity (MRA)

L130 included into the Bio-Rad buffer “change with” included in the Bio-Rad buffer

L132 immobilized; L141 analyse; L143 normalized. Please check all manuscript choose North American or British spellings and be consistent thorough all text.

L134 were used to perform first “change with” performed first

L135 electrophoresis respectively. “change with” electrophoresis, respectively.

L139 scanned with the using of an “change with” scanned using an

L171 The establishment of the mixed models which were prediction models for muscle colour traits was based on the method of Starkey et al. [26]. “change with” The mixed models, which were prediction models for muscle colour traits, were based on Starkey et al. [26].

L183 Datas were expressed “change with” Data were expressed

L188 kyoto encyclopedia “change with” Kyoto encyclopedia

L201-202 Besides being as a colour parameter, the L*-value also considered to be an indicator of water holding capacity of meat [11]. “change with” Besides being a colour parameter, the L*-value is also considered an indicator of the water holding capacity of meat [11].

L204 considered to be one “change with” considered one

L211 partially agreement “change with” partially in agreement

L222 MRA also decreased with storage time “change with” MRA also decreased as storage time

L228 and reduction material consumption “change with” and reduction in material consumption

Table 1 Please check “Means in rows with different superscripts (a-c, x-z) are different (P < 0.05).” Maybe the authors want convey “Means in rows with different superscripts (a-c) are different (P < 0.05) and means in column for LL and PM for each storage time and attribute with different superscripts (x-z) are different (P < 0.05).”

Please change LD with LL in Table 1. Please check all manuscript for similar issues

L239 of spots was at “change with” of spots were at

L240 information were shown in Table 2 “change with” information are shown in Table 2

L263 method of pearson's correlation “change with” method of Pearson's correlation

L265 traits which were affected “change with” traits that were affected

L281 were significantly “change with” that were significantly

L299 observed in during late stage “change with” observed during the late stage

L301 glycolysis metablic “change with” glycolysis metabolic

L318 of above-mentioned enzymes “change with” of the enzymes mentioned above

L322 Besides, NADH which supplied by glycolysis is necessary for NADH–cytochrome b5 reductase system of mitochondria [41]. “change with” Besides, NADH supplied by glycolysis is necessary for the NADH–cytochrome b5 reductase system of mitochondria [41].

L326 Researchers [31] considered NADH as a critical component of MRA, whose metabolic function plays a key role in reducing the accumulation of metmyoglobin [19]. “change with” Researchers [31] considered NADH a critical component of MRA, whose metabolic function plays a key role in reducing metmyoglobin accumulation [19].

L345 It provides a way to catalyze the conversion of ADP to ATP, as well as to convert AMP to ADP in the cytoplasm [46]. “change with” It provides a way to catalyze the conversion of ADP to ATP and convert AMP to ADP in the cytoplasm [46].

L364 Creatine kinase M-type which is relatively overabundant can “change with” Creatine kinase M-type, which is relatively overabundant, can

L367 anhydrase 3 was considered “change with” anhydrase 3 were considered

L369 in M. longissimuss “change with” in M. longissimus

L392 been identified related to “change with” been identified as related to

L393 proteomic analysis “change with” proteomic analyses

L394 chain 1 which were identified in porcine longissimus muscle were related “change with” chain 1, identified in porcine longissimus muscle were related

L397 great interest for discriminating “change with” great interest in discriminating

L413 proteins interacted strongly “change with” proteins that interacted strongly

L413 with each others “change with” with each other

L444 This result was consistent with the related researches [2] whose interpretation of possible mechanism as follows. “change with” This result was consistent with the related research [2] whose interpretation of possible mechanisms is as follows.

L453 proteome proiles associated “change with” proteome profiles associated

L463 in glycolysis pathway “change with” in the glycolysis pathway

L466 potential probability to be predictors “change with” potential probability of being predictors

Author Response

Article Title: Proteomic Changes in Sarcoplasmic and Myofibrillar Proteins associated with Colour Stability of Ovine Muscle during Post-mortem Storage

Dear reviewers of Foods,

Thank you very much for your supervision of our manuscript. The critical comments and thoughtful suggestions from you not only helped us with the improvement of our manuscript, but also provided some enlightening ideas for our future research. Based on these comments and suggestions, we have made careful modifications on the original manuscript. Responses to each comment have been addressed as below.

We are looking forward to hear from you.

Sincerely yours,

Ruitong Dai on behalf of the authors

Ruitong Dai 

College of Food Science and Nutritional Engineering, China Agricultural University

17 Qinghua East Road,100083, Beijing, China

E-mail address: dairuitong@hotmail.com

Tel.: 86-10-62737547, Fax: 86-10-62839300

First author:

Xiaoguang Gao1, 2

1 College of Bioscience and Bioengineering, Hebei University of Science and Technology, No.26 Yuxiang Street, Yuhua District, Shijiazhuang, Hebei, 050000, P. R. China

2 College of Food Science and Nutritional Engineering, China Agricultural University, No.17 Qinghua East Road, Haidian District, Beijing, 100083, P. R. China

E-mail address: gaoxiaoguang23@hotmail.com

Responses and changes based on reviewer(s)' comments:

Comments and Suggestions for Authors

Proteomic Changes in Sarcoplasmic and Myofibrillar Proteins associated with Colour Stability of Ovine Muscle during Post-mortem Storage

This article represents a significant contribution to scientific knowledge about the proteomic characteristics for the sarcoplasmic and myofibrillar proteome of two lamb muscles (M. longissimus lumborum and M. psoas major) and their relationships with muscle colour-stable and colour-labile.

The article is well organized and written clearly. Despite this, some improvements can be made to make the text clearer. The text addresses the subject with scientific correction. Tables and figures are relevant, but some corrections are needed to improve accuracy and clarity. The material and methods are clearly described, which will allow a perfect understanding by other researchers. The results are well discussed with the existing knowledge on the subject. Finally, the results support the conclusions.

Some detailed comments below:

L30 always been connected “change with” have always been connected

Response

Thank you very much for your careful check. It has been corrected according to the reviewer's instruction.

L33 for meat industry “change with” for the meat industry

Response

Thank you very much for your careful check. It has been corrected according to the reviewer’s instruction.

L49 development including the development, including the

Response

Thank you very much for your careful check. It has been corrected according to the reviewer’s instruction.

L60-61The samples of skeletal muscle were harvested from the Small-tailed Han Sheep (male) were slaughtered at the age of 8 months and mean carcass weight of 15.62 kg “change with” The skeletal muscle samples were harvested from 6 male Small-tailed Han Sheep slaughtered at eight months with a mean carcass weight of 15.6± kg. Please introduce the standard deviation of the carcass weight.

Response

Thank you very much for your careful check. It has been corrected according to the reviewer’s instruction. The standard deviation of the carcass weight was “15.6±0.2 kg”.

L63-64. Improve the description text of the slaughter procedure and introduce a sentence related with animal welfare.

Response

Thank you very much for your careful check and enlightening reminding. According to the reviewer's instruction, this part has been revised as follows:

The animals received humanely nonpainful manipulations at the termination of the procedure without regaining consciousness

L64 slaughter was following the industrial “change with” slaughter followed the industrial

Response

Thank you very much for your careful check. It has been corrected according to the reviewer’s instruction.

L70 and stored in a in a refrigerator for storage time of five days “change with” and put in a refrigerator for a storage time of five days

Response

Thank you very much for your careful check. It has been corrected according to the reviewer’s instruction.

L71 Muscles were taken out to analyse at day 1, day 3 and day 5 for further experiments. “change with” Muscles were taken out to analyse at day 1, day 3 and day 5.

Response

Thank you very much for your careful check. It has been corrected according to the reviewer’s instruction.

L76 Please check the word accordingly. Please rewrite this sentence.

Response

Thank you very much for your careful check and enlightening reminding. According to the reviewer's instruction, this sentence has been re written as follows:

At the end of storage period, myofibrillar proteins were extracted for the comparison of proteomic difference between muscle types (LL and PM).

L79 determined for the surface of each “change with” determined on the surface of each

Response

Thank you very much for your careful check. It has been corrected according to the reviewer’s instruction.

L91 Absorbance of filtrate was tested “change with” The absorbance of the filtrate was tested

Response

Thank you very much for your careful check. It has been corrected according to the reviewer’s instruction.

L99 MRA (metmyoglobin reducing activity) “change with” The metmyoglobin reducing activity (MRA)

Response

Thank you very much for your careful check. It has been corrected according to the reviewer’s instruction.

L130 included into the Bio-Rad buffer “change with” included in the Bio-Rad buffer

Response

Thank you very much for your careful check. It has been corrected according to the reviewer’s instruction.

L132 immobilized; L141 analyse; L143 normalized. Please check all manuscript choose North American or British spellings and be consistent thorough all text.

Response

Thank you very much for your careful check. This problem has been corrected thorough all text according to the reviewer’s instruction.

L134 were used to perform first “change with” performed first

Response

Thank you very much for your careful check. It has been corrected according to the reviewer’s instruction.

L135 electrophoresis respectively. “change with” electrophoresis, respectively.

Response

Thank you very much for your careful check. It has been corrected according to the reviewer’s instruction.

L139 scanned with the using of an “change with” scanned using an

Response

Thank you very much for your careful check. It has been corrected according to the reviewer’s instruction.

L171 The establishment of the mixed models which were prediction models for muscle colour traits was based on the method of Starkey et al. [26]. “change with” The mixed models, which were prediction models for muscle colour traits, were based on Starkey et al. [26].

Response

Thank you very much for your careful check. It has been corrected according to the reviewer’s instruction.

According to Academic Editor's Comments, Table 4 and the related discussion has been removed. Therefore, the sentence has been eventually deleted. (Please see another word document named “Responses to Academic Editor's Comments”)

L183 Datas were expressed “change with” Data were expressed

Response

Thank you very much for your careful check. It has been corrected according to the reviewer’s instruction.

L188 kyoto encyclopedia “change with” Kyoto encyclopedia

Response

Thank you very much for your careful check. It has been corrected according to the reviewer’s instruction.

L201-202 Besides being as a colour parameter, the L*-value also considered to be an indicator of water holding capacity of meat [11]. “change with” Besides being a colour parameter, the L*-value is also considered an indicator of the water holding capacity of meat [11].

Response

Thank you very much for your careful check. It has been corrected according to the reviewer’s instruction.

L204 considered to be one “change with” considered one

Response

Thank you very much for your careful check. It has been corrected according to the reviewer’s instruction.

L211 partially agreement “change with” partially in agreement

Response

Thank you very much for your careful check. It has been corrected according to the reviewer’s instruction.

L222 MRA also decreased with storage time “change with” MRA also decreased as storage time

Response

Thank you very much for your careful check. It has been corrected according to the reviewer’s instruction.

L228 and reduction material consumption “change with” and reduction in material consumption

Response

Thank you very much for your careful check. It has been corrected according to the reviewer’s instruction.

Table 1 Please check “Means in rows with different superscripts (a-c, x-z) are different (P < 0.05).” Maybe the authors want convey “Means in rows with different superscripts (a-c) are different (P < 0.05) and means in column for LL and PM for each storage time and attribute with different superscripts (x-z) are different (P < 0.05).”

Response

Thank you very much for your careful check. We are very sorry for this error. It has been corrected according to the reviewer’s instruction.

Please change LD with LL in Table 1. Please check all manuscript for similar issues

Response

Thank you very much for your careful check. We are very sorry for this error. It has been corrected according to the reviewer’s instruction.

L239 of spots was at “change with” of spots were at

Response

Thank you very much for your careful check. We are very sorry for this error. It has been corrected according to the reviewer’s instruction.

L240 information were shown in Table 2 “change with” information are shown in Table 2

Response

Thank you very much for your careful check. We are very sorry for this error. It has been corrected according to the reviewer’s instruction.

L263 method of pearson's correlation “change with” method of Pearson's correlation

Response

Thank you very much for your careful check. It has been corrected according to the reviewer’s instruction.

L265 traits which were affected “change with” traits that were affected

Response

Thank you very much for your careful check. It has been corrected according to the reviewer’s instruction.

L281 were significantly “change with” that were significantly

Response

Thank you very much for your careful check. It has been corrected according to the reviewer’s instruction.

L299 observed in during late stage “change with” observed during the late stage

Response

Thank you very much for your careful check. It has been corrected according to the reviewer’s instruction.

L301 glycolysis metablic “change with” glycolysis metabolic

Response

Thank you very much for your careful check. It has been corrected according to the reviewer’s instruction.

L318 of above-mentioned enzymes “change with” of the enzymes mentioned above

Response

Thank you very much for your careful check. It has been corrected according to the reviewer’s instruction.

L322 Besides, NADH which supplied by glycolysis is necessary for NADH–cytochrome b5 reductase system of mitochondria [41]. “change with” Besides, NADH supplied by glycolysis is necessary for the NADH–cytochrome b5 reductase system of mitochondria [41].

Response

Thank you very much for your careful check. It has been corrected according to the reviewer’s instruction.

L326 Researchers [31] considered NADH as a critical component of MRA, whose metabolic function plays a key role in reducing the accumulation of metmyoglobin [19]. “change with” Researchers [31] considered NADH a critical component of MRA, whose metabolic function plays a key role in reducing metmyoglobin accumulation [19].

Response

Thank you very much for your careful check. It has been corrected according to the reviewer’s instruction.

L345 It provides a way to catalyze the conversion of ADP to ATP, as well as to convert AMP to ADP in the cytoplasm [46]. “change with” It provides a way to catalyze the conversion of ADP to ATP and convert AMP to ADP in the cytoplasm [46].

Response

Thank you very much for your careful check. It has been corrected according to the reviewer’s instruction.

L364 Creatine kinase M-type which is relatively overabundant can “change with” Creatine kinase M-type, which is relatively overabundant, can

Response

Thank you very much for your careful check. It has been corrected according to the reviewer’s instruction.

L367 anhydrase 3 was considered “change with” anhydrase 3 were considered

Response

Thank you very much for your careful check. It has been corrected according to the reviewer’s instruction.

L369 in M. longissimuss “change with” in M. Longissimus

Response

Thank you very much for your careful check. It has been corrected according to the reviewer’s instruction.

L392 been identified related to “change with” been identified as related to

Response

Thank you very much for your careful check. It has been corrected according to the reviewer’s instruction.

L393 proteomic analysis “change with” proteomic analyses

Response

Thank you very much for your careful check. It has been corrected according to the reviewer’s instruction.

L394 chain 1 which were identified in porcine longissimus muscle were related “change with” chain 1, identified in porcine longissimus muscle were related

Response

Thank you very much for your careful check. It has been corrected according to the reviewer’s instruction.

L397 great interest for discriminating “change with” great interest in discriminating

Response

Thank you very much for your careful check. It has been corrected according to the reviewer’s instruction.

L413 proteins interacted strongly “change with” proteins that interacted strongly

Response

Thank you very much for your careful check. It has been corrected according to the reviewer’s instruction.

L413 with each others “change with” with each other

Response

Thank you very much for your careful check. It has been corrected according to the reviewer’s instruction.

L444 This result was consistent with the related researches [2] whose interpretation of possible mechanism as follows. “change with” This result was consistent with the related research [2] whose interpretation of possible mechanisms is as follows.

Response

Thank you very much for your careful check. It has been corrected according to the reviewer’s instruction.

L453 proteome proiles associated “change with” proteome profiles associated

Response

Thank you very much for your careful check. It has been corrected according to the reviewer’s instruction.

L463 in glycolysis pathway “change with” in the glycolysis pathway

Response

Thank you very much for your careful check. It has been corrected according to the reviewer’s instruction.

L466 potential probability to be predictors “change with” potential probability of being predictors

Response

Thank you very much for your careful check. It has been corrected according to the reviewer’s instruction. 

Supplementary content:

In the PDF version of the manuscript the reviewer mentioned that:

L221 Figure 1 dose not show PM. Please explain the strips and color of the bars

Response

Thank you very much for your careful check. Originally, the version of the picture we uploaded was correct. The legend display may be incomplete due to compatibility problems. Now, it has been corrected according to the reviewer’s instruction. 

L230 Table 1 is mentioned in the text before Figure1. It should be displayed first too.

Response

Thank you very much for your careful check. It has been corrected according to the reviewer’s instruction. 

AND

Responses and changes based on editor(s)' comments:

Academic Editor Comments for Author

Academic Editor Comments

Table 4, should be removed. The models as they are are biased and not accurate. Be careful of reproducing some errors of the large literature. The number of animals doesn't allow such models. The factors were mixed, thus better to avoid any confusion.

Response

Thank you very much for your careful check and enlightening reminding. According to the editor's instruction, Table 4 and the related discussion (and two related references) has been removed.

Reviewer 2 Report

The research presented in this manuscript is highly interesting, well structured and sound scientifically. Review of the English grammar is required. Comments are in the text.

Author Response

Article Title: Proteomic Changes in Sarcoplasmic and Myofibrillar Proteins associated with Colour Stability of Ovine Muscle during Post-mortem Storage

Dear reviewers of Foods,

Thank you very much for your supervision of our manuscript. The critical comments and thoughtful suggestions from you not only helped us with the improvement of our manuscript, but also provided some enlightening ideas for our future research. Based on these comments and suggestions, we have made careful modifications on the original manuscript. Responses to each comment have been addressed as below.

We are looking forward to hear from you.

Sincerely yours,

Ruitong Dai on behalf of the authors

Ruitong Dai 

College of Food Science and Nutritional Engineering, China Agricultural University

17 Qinghua East Road,100083, Beijing, China

E-mail address: dairuitong@hotmail.com

Tel.: 86-10-62737547, Fax: 86-10-62839300

First author:

Xiaoguang Gao1, 2

1 College of Bioscience and Bioengineering, Hebei University of Science and Technology, No.26 Yuxiang Street, Yuhua District, Shijiazhuang, Hebei, 050000, P. R. China

2 College of Food Science and Nutritional Engineering, China Agricultural University, No.17 Qinghua East Road, Haidian District, Beijing, 100083, P. R. China

E-mail address: gaoxiaoguang23@hotmail.com

Responses and changes based on reviewer(s)' comments:

Comments and Suggestions for Authors

Proteomic Changes in Sarcoplasmic and Myofibrillar Proteins associated with Colour Stability of Ovine Muscle during Post-mortem Storage

This article represents a significant contribution to scientific knowledge about the proteomic characteristics for the sarcoplasmic and myofibrillar proteome of two lamb muscles (M. longissimus lumborum and M. psoas major) and their relationships with muscle colour-stable and colour-labile.

The article is well organized and written clearly. Despite this, some improvements can be made to make the text clearer. The text addresses the subject with scientific correction. Tables and figures are relevant, but some corrections are needed to improve accuracy and clarity. The material and methods are clearly described, which will allow a perfect understanding by other researchers. The results are well discussed with the existing knowledge on the subject. Finally, the results support the conclusions.

Some detailed comments below:

L30 always been connected “change with” have always been connected

Response

Thank you very much for your careful check. It has been corrected according to the reviewer's instruction.

L33 for meat industry “change with” for the meat industry

Response

Thank you very much for your careful check. It has been corrected according to the reviewer’s instruction.

L49 development including the development, including the

Response

Thank you very much for your careful check. It has been corrected according to the reviewer’s instruction.

L60-61The samples of skeletal muscle were harvested from the Small-tailed Han Sheep (male) were slaughtered at the age of 8 months and mean carcass weight of 15.62 kg “change with” The skeletal muscle samples were harvested from 6 male Small-tailed Han Sheep slaughtered at eight months with a mean carcass weight of 15.6± kg. Please introduce the standard deviation of the carcass weight.

Response

Thank you very much for your careful check. It has been corrected according to the reviewer’s instruction. The standard deviation of the carcass weight was “15.6±0.2 kg”.

L63-64. Improve the description text of the slaughter procedure and introduce a sentence related with animal welfare.

Response

Thank you very much for your careful check and enlightening reminding. According to the reviewer's instruction, this part has been revised as follows:

The animals received humanely nonpainful manipulations at the termination of the procedure without regaining consciousness.

L64 slaughter was following the industrial “change with” slaughter followed the industrial

Response

Thank you very much for your careful check. It has been corrected according to the reviewer’s instruction.

L70 and stored in a in a refrigerator for storage time of five days “change with” and put in a refrigerator for a storage time of five days

Response

Thank you very much for your careful check. It has been corrected according to the reviewer’s instruction.

L71 Muscles were taken out to analyse at day 1, day 3 and day 5 for further experiments. “change with” Muscles were taken out to analyse at day 1, day 3 and day 5.

Response

Thank you very much for your careful check. It has been corrected according to the reviewer’s instruction.

L76 Please check the word accordingly. Please rewrite this sentence.

Response

Thank you very much for your careful check and enlightening reminding. According to the reviewer's instruction, this sentence has been re written as follows:

At the end of storage period, myofibrillar proteins were extracted for the comparison of proteomic difference between muscle types (LL and PM).

L79 determined for the surface of each “change with” determined on the surface of each

Response

Thank you very much for your careful check. It has been corrected according to the reviewer’s instruction.

L91 Absorbance of filtrate was tested “change with” The absorbance of the filtrate was tested

Response

Thank you very much for your careful check. It has been corrected according to the reviewer’s instruction.

L99 MRA (metmyoglobin reducing activity) “change with” The metmyoglobin reducing activity (MRA)

Response

Thank you very much for your careful check. It has been corrected according to the reviewer’s instruction.

L130 included into the Bio-Rad buffer “change with” included in the Bio-Rad buffer

Response

Thank you very much for your careful check. It has been corrected according to the reviewer’s instruction.

L132 immobilized; L141 analyse; L143 normalized. Please check all manuscript choose North American or British spellings and be consistent thorough all text.

Response

Thank you very much for your careful check. This problem has been corrected thorough all text according to the reviewer’s instruction.

L134 were used to perform first “change with” performed first

Response

Thank you very much for your careful check. It has been corrected according to the reviewer’s instruction.

L135 electrophoresis respectively. “change with” electrophoresis, respectively.

Response

Thank you very much for your careful check. It has been corrected according to the reviewer’s instruction.

L139 scanned with the using of an “change with” scanned using an

Response

Thank you very much for your careful check. It has been corrected according to the reviewer’s instruction.

L171 The establishment of the mixed models which were prediction models for muscle colour traits was based on the method of Starkey et al. [26]. “change with” The mixed models, which were prediction models for muscle colour traits, were based on Starkey et al. [26].

Response

Thank you very much for your careful check. It has been corrected according to the reviewer’s instruction.

According to Academic Editor's Comments, Table 4 and the related discussion has been removed. Therefore, the sentence has been eventually deleted. (Please see another word document named “Responses to Academic Editor's Comments”)

L183 Datas were expressed “change with” Data were expressed

Response

Thank you very much for your careful check. It has been corrected according to the reviewer’s instruction.

L188 kyoto encyclopedia “change with” Kyoto encyclopedia

Response

Thank you very much for your careful check. It has been corrected according to the reviewer’s instruction.

L201-202 Besides being as a colour parameter, the L*-value also considered to be an indicator of water holding capacity of meat [11]. “change with” Besides being a colour parameter, the L*-value is also considered an indicator of the water holding capacity of meat [11].

Response

Thank you very much for your careful check. It has been corrected according to the reviewer’s instruction.

L204 considered to be one “change with” considered one

Response

Thank you very much for your careful check. It has been corrected according to the reviewer’s instruction.

L211 partially agreement “change with” partially in agreement

Response

Thank you very much for your careful check. It has been corrected according to the reviewer’s instruction.

L222 MRA also decreased with storage time “change with” MRA also decreased as storage time

Response

Thank you very much for your careful check. It has been corrected according to the reviewer’s instruction.

L228 and reduction material consumption “change with” and reduction in material consumption

Response

Thank you very much for your careful check. It has been corrected according to the reviewer’s instruction.

Table 1 Please check “Means in rows with different superscripts (a-c, x-z) are different (P < 0.05).” Maybe the authors want convey “Means in rows with different superscripts (a-c) are different (P < 0.05) and means in column for LL and PM for each storage time and attribute with different superscripts (x-z) are different (P < 0.05).”

Response

Thank you very much for your careful check. We are very sorry for this error. It has been corrected according to the reviewer’s instruction.

Please change LD with LL in Table 1. Please check all manuscript for similar issues

Response

Thank you very much for your careful check. We are very sorry for this error. It has been corrected according to the reviewer’s instruction.

L239 of spots was at “change with” of spots were at

Response

Thank you very much for your careful check. We are very sorry for this error. It has been corrected according to the reviewer’s instruction.

L240 information were shown in Table 2 “change with” information are shown in Table 2

Response

Thank you very much for your careful check. We are very sorry for this error. It has been corrected according to the reviewer’s instruction.

L263 method of pearson's correlation “change with” method of Pearson's correlation

Response

Thank you very much for your careful check. It has been corrected according to the reviewer’s instruction.

L265 traits which were affected “change with” traits that were affected

Response

Thank you very much for your careful check. It has been corrected according to the reviewer’s instruction.

L281 were significantly “change with” that were significantly

Response

Thank you very much for your careful check. It has been corrected according to the reviewer’s instruction.

L299 observed in during late stage “change with” observed during the late stage

Response

Thank you very much for your careful check. It has been corrected according to the reviewer’s instruction.

L301 glycolysis metablic “change with” glycolysis metabolic

Response

Thank you very much for your careful check. It has been corrected according to the reviewer’s instruction.

L318 of above-mentioned enzymes “change with” of the enzymes mentioned above

Response

Thank you very much for your careful check. It has been corrected according to the reviewer’s instruction.

L322 Besides, NADH which supplied by glycolysis is necessary for NADH–cytochrome b5 reductase system of mitochondria [41]. “change with” Besides, NADH supplied by glycolysis is necessary for the NADH–cytochrome b5 reductase system of mitochondria [41].

Response

Thank you very much for your careful check. It has been corrected according to the reviewer’s instruction.

L326 Researchers [31] considered NADH as a critical component of MRA, whose metabolic function plays a key role in reducing the accumulation of metmyoglobin [19]. “change with” Researchers [31] considered NADH a critical component of MRA, whose metabolic function plays a key role in reducing metmyoglobin accumulation [19].

Response

Thank you very much for your careful check. It has been corrected according to the reviewer’s instruction.

L345 It provides a way to catalyze the conversion of ADP to ATP, as well as to convert AMP to ADP in the cytoplasm [46]. “change with” It provides a way to catalyze the conversion of ADP to ATP and convert AMP to ADP in the cytoplasm [46].

Response

Thank you very much for your careful check. It has been corrected according to the reviewer’s instruction.

L364 Creatine kinase M-type which is relatively overabundant can “change with” Creatine kinase M-type, which is relatively overabundant, can

Response

Thank you very much for your careful check. It has been corrected according to the reviewer’s instruction.

L367 anhydrase 3 was considered “change with” anhydrase 3 were considered

Response

Thank you very much for your careful check. It has been corrected according to the reviewer’s instruction.

L369 in M. longissimuss “change with” in M. Longissimus

Response

Thank you very much for your careful check. It has been corrected according to the reviewer’s instruction.

L392 been identified related to “change with” been identified as related to

Response

Thank you very much for your careful check. It has been corrected according to the reviewer’s instruction.

L393 proteomic analysis “change with” proteomic analyses

Response

Thank you very much for your careful check. It has been corrected according to the reviewer’s instruction.

L394 chain 1 which were identified in porcine longissimus muscle were related “change with” chain 1, identified in porcine longissimus muscle were related

Response

Thank you very much for your careful check. It has been corrected according to the reviewer’s instruction.

L397 great interest for discriminating “change with” great interest in discriminating

Response

Thank you very much for your careful check. It has been corrected according to the reviewer’s instruction.

L413 proteins interacted strongly “change with” proteins that interacted strongly

Response

Thank you very much for your careful check. It has been corrected according to the reviewer’s instruction.

L413 with each others “change with” with each other

Response

Thank you very much for your careful check. It has been corrected according to the reviewer’s instruction.

L444 This result was consistent with the related researches [2] whose interpretation of possible mechanism as follows. “change with” This result was consistent with the related research [2] whose interpretation of possible mechanisms is as follows.

Response

Thank you very much for your careful check. It has been corrected according to the reviewer’s instruction.

L453 proteome proiles associated “change with” proteome profiles associated

Response

Thank you very much for your careful check. It has been corrected according to the reviewer’s instruction.

L463 in glycolysis pathway “change with” in the glycolysis pathway

Response

Thank you very much for your careful check. It has been corrected according to the reviewer’s instruction.

L466 potential probability to be predictors “change with” potential probability of being predictors

Response

Thank you very much for your careful check. It has been corrected according to the reviewer’s instruction. 

Supplementary content:

In the PDF version of the manuscript the reviewer mentioned that:

L221 Figure 1 dose not show PM. Please explain the strips and color of the bars

Response

Thank you very much for your careful check. Originally, the version of the picture we uploaded was correct. The legend display may be incomplete due to compatibility problems. Now, it has been corrected according to the reviewer’s instruction. 

L230 Table 1 is mentioned in the text before Figure1. It should be displayed first too.

Response

Thank you very much for your careful check. It has been corrected according to the reviewer’s instruction. 

AND

Responses and changes based on editor(s)' comments:

Academic Editor Comments for Author

Academic Editor Comments

Table 4, should be removed. The models as they are are biased and not accurate. Be careful of reproducing some errors of the large literature. The number of animals doesn't allow such models. The factors were mixed, thus better to avoid any confusion.

Response

Thank you very much for your careful check and enlightening reminding. According to the editor's instruction, Table 4 and the related discussion (and two related references) has been removed.
